# Energy-Inspired Models: Learning with Sampler-Induced Distributions

**Dieterich Lawson**[*][†]
Stanford University
jdlawson@stanford.edu

**George Tucker**[*]**, Bo Dai**
Google Research, Brain Team
{gjt, bodai}@google.com

**Rajesh Ranganath**
New York University
rajeshr@cims.nyu.edu

## Abstract

Energy-based models (EBMs) are powerful probabilistic models [8, 44], but suffer from intractable sampling and density evaluation due to the partition function. As a result, inference in EBMs relies on approximate sampling algorithms, leading to a mismatch between the model and inference. Motivated by this, we consider the sampler-induced distribution as the model of interest and maximize the likelihood of this model. This yields a class of *energy-inspired models* (EIMs) that incorporate learned energy functions while still providing exact samples and tractable log-likelihood lower bounds. We describe and evaluate three instantiations of such models based on truncated rejection sampling, self-normalized importance sampling, and Hamiltonian importance sampling. These models outperform or perform comparably to the recently proposed Learned Accept/Reject Sampling algorithm [5] and provide new insights on ranking Noise Contrastive Estimation [34, 46] and Contrastive Predictive Coding [57]. Moreover, EIMs allow us to generalize a recent connection between multi-sample variational lower bounds [9] and auxiliary variable variational inference [1, 63, 59, 47]. We show how recent variational bounds [9, 49, 52, 42, 73, 51, 65] can be unified with EIMs as the variational family.

## 1  Introduction

Energy-based models (EBMs) have a long history in statistics and machine learning [16, 75, 44]. EBMs score configurations of variables with an energy function, which induces a distribution on the variables in the form of a Gibbs distribution. Different choices of energy function recover well-known probabilistic models including Markov random fields [36], (restricted) Boltzmann machines [64, 24, 30], and conditional random fields [41]. However, this flexibility comes at the cost of challenging inference and learning: both sampling and density evaluation of EBMs are generally intractable, which hinders the applications of EBMs in practice.

Because of the intractability of general EBMs, practical implementations rely on approximate sampling procedures (*e.g.*, Markov chain Monte Carlo (MCMC)) for inference. This creates a mismatch between the model and the approximate inference procedure, and can lead to suboptimal performance and unstable training when approximate samples are used in the training procedure.

Currently, most attempts to fix the mismatch lie in designing better sampling algorithms (*e.g.*, Hamiltonian Monte Carlo [54], annealed importance sampling [53]) or exploiting variational techniques [35, 15, 14] to reduce the inference approximation error.

---

[*]Equal contributions. [†]Research performed while at New York University.
Code and image samples: `sites.google.com/view/energy-inspired-models`.

Instead, we bridge the gap between the model and inference by directly treating the sampling procedure as the model of interest and optimizing the log-likelihood of the the sampling procedure. We call these models *energy-inspired models* (EIMs) because they incorporate a learned energy function while providing tractable, exact samples. This shift in perspective aligns the training and sampling procedure, leading to principled and consistent training and inference.

To accomplish this, we cast the sampling procedure as a latent variable model. This allows us to maximize variational lower bounds [33, 7] on the log-likelihood (c.f., Kingma and Welling [38], Rezende et al. [61]). To illustrate this, we develop and evaluate energy-inspired models based on truncated rejection sampling (Algorithm 1), self-normalized importance sampling (Algorithm 2), and Hamiltonian importance sampling (Algorithm 3). Interestingly, the model based on self-normalized importance sampling is closely related to *ranking* NCE [34, 46], suggesting a principled objective for training the "noise" distribution.

Our second contribution is to show that EIMs provide a unifying conceptual framework to explain many advances in constructing tighter variational lower bounds for latent variable models (*e.g.*, [9, 49, 52, 42, 73, 51, 65]). Previously, each bound required a separate derivation and evaluation, and their relationship was unclear. We show that these bounds can be viewed as specific instances of auxiliary variable variational inference [1, 63, 59, 47] with different EIMs as the variational family. Based on general results for auxiliary latent variables, this immediately gives rise to a variational lower bound with a characterization of the tightness of the bound. Furthermore, this unified view highlights the implicit (potentially suboptimal) choices made and exposes the reusable components that can be combined to form novel variational lower bounds. Concurrently, Domke and Sheldon [19] note a similar connection, however, their focus is on the use of the variational distribution for posterior inference.

In summary, our contributions are:

- The construction of a tractable class of *energy-inspired models* (EIMs), which lead to *consistent* learning and inference. To illustrate this, we build models with truncated rejection sampling, self-normalized importance sampling, and Hamiltonian importance sampling and evaluate them on synthetic and real-world tasks. These models can be fit by maximizing a tractable lower bound on their log-likelihood.

- We show that EIMs with auxiliary variable variational inference provide a unifying framework for understanding recent tighter variational lower bounds, simplifying their analysis and exposing potentially sub-optimal design choices.

## 2   Background

In this work, we consider learned probabilistic models of data $p(x)$. Energy-based models [44] define $p(x)$ in terms of an energy function $U(x)$

$$p(x) = \frac{\pi(x) \exp(-U(x))}{Z},$$

where $\pi$ is a tractable "prior" distribution and $Z = \int \pi(x) \exp(-U(x)) \, dx$ is a generally intractable partition function. To fit the model, many approximate methods have been developed (*e.g.*, pseudo log-likelihood [6], contrastive divergence [30, 67], score matching estimator [31], minimum probability flow [66], noise contrastive estimation [28]) to bypass the calculation of the partition function. Empirically, previous work has found that convolutional architectures that score images (*i.e.*, map $x$ to a real number) tend to have strong inductive biases that match natural data (e.g., [70, 71, 72, 25, 22]). These networks are a natural fit for energy-based models. Because drawing exact samples from these models is intractable, samples are typically approximated by Monte Carlo schemes, for example, Hamiltonian Monte Carlo [55].

Alternatively, latent variables $z$ allow us to construct complex distributions by defining the likelihood $p(x) = \int p(x|z)p(z) \, dz$ in terms of tractable components $p(z)$ and $p(x|z)$. While marginalizing $z$ is generally intractable, we can instead optimize a tractable lower bound on $\log p(x)$ using the identity

$$\log p(x) = \mathbb{E}_{q(z|x)} \left[ \log \frac{p(x,z)}{q(z|x)} \right] + D_{\mathrm{KL}} \left( q(z|x) || p(z|x) \right), \tag{1}$$

where $q(z|x)$ is a variational distribution and the positive $D_{\mathrm{KL}}$ term can be omitted to form a lower bound commonly referred to as the evidence lower bound (ELBO) [33, 7]. The tightness of the bound is controlled by how accurately $q(z|x)$ models $p(z|x)$, so limited expressivity in the variational family can negatively impact the learned model.

## 3 Energy-Inspired Models

Instead of viewing the sampling procedure as drawing approximate samples from the energy-based models, we treat the sampling procedure as the model of interest. We represent the randomness in the sampler as latent variables, and we obtain a tractable lower bound on the marginal likelihood using the ELBO. Explicitly, if $p(\lambda)$ represents the randomness in the sampler and $p(x|\lambda)$ is the generative process, then

$$\log p(x) \geq \mathbb{E}_{q(\lambda|x)} \left[ \log \frac{p(\lambda)p(x|\lambda)}{q(\lambda|x)} \right], \tag{2}$$

where $q(\lambda|x)$ is a variational distribution that can be optimized to tighten the bound. In this section, we explore concrete instantiations of models in this paradigm: one based on truncated rejection sampling (TRS), one based on self-normalized importance sampling (SNIS), and another based on Hamiltonian importance sampling (HIS) [54].

---

**Algorithm 1** TRS$(\pi, U, T)$ generative process

---

**Require:** Proposal distribution $\pi(x)$, energy function $U(x)$, and truncation step $T$.
 1: **for** $t = 1, \ldots, T - 1$ **do**
 2:     Sample $x_t \sim \pi(x)$.
 3:     Sample $b_t \sim \text{Bernoulli}(\sigma(-U(x_t)))$.
 4: **end for**
 5: Sample $x_T \sim \pi(x)$ and set $b_T = 1$.
 6: Compute $i = \min t$ s.t. $b_t = 1$.
 7: **return** $x = x_i$.

---

### 3.1 Truncated Rejection Sampling (TRS)

Consider the truncated rejection sampling process (Algorithm 1) used in [5], where we sequentially draw a sample $x_t$ from $\pi(x)$ and accept it with probability $\sigma(-U(x_t))$. To ensure that the process ends, if we have not accepted a sample after $T$ steps, then we return $x_T$.

In this case, $\lambda = (x_{1:T}, b_{1:T-1}, i)$, so we need to construct a variational distribution $q(\lambda|x)$. The optimal $q(\lambda|x)$ is $p(\lambda|x)$, which motivates choosing a similarly structured variational distribution. It is straightforward to see that $p(i|x) \propto (1 - Z)^{i-1}\sigma(-U(x))^{\delta_i < T}$, where $Z = \int \pi(x)\sigma(-U(x))\,dx$ is generally intractable. So, we choose $q(i|x) \propto (1 - \hat{Z})^{i-1}\sigma(-U(x))^{\delta_i < T}$, where $\hat{Z}$ is a learnable variational parameter. Then, we sample $x_{1:T}$ and $b_{i+1:T-1}$ as in the generative process. This results in a simple variational bound

$$\log p_{TRS}(x) \geq \mathbb{E}_{q(i|x)}\mathbb{E}_{\prod_{t=1}^{i} \pi(x_t)} \left[ \log \pi(x)\sigma(-U(x)) + \sum_{t=1}^{i-1} \log\left(1 - \sigma(-U(x_t))\right) - \log q(i|x) \right].$$

The TRS generative process is the same process as the Learned Accept/Reject Sampling (LARS) model [5]. The key difference is the training procedure. LARS tries to directly estimate the gradient of the log likelihood. Without truncation, such a process is attractive because unbiased gradients of its log likelihood can easily be computed without knowing the normalizing constant. Unfortunately, after truncating the process, we require estimating a normalizing constant. In practice, Bauer and Mnih [5] estimate the normalizing constant using $1024$ samples during training and $10^{10}$ samples during evaluation. Even so, LARS requires additional implementation tricks (*e.g.*, evaluating the target density, using an exponential moving average to estimate the normalizing constant) to ensure successful training, which complicate the implementation and analysis of the algorithm. On the other hand, we optimize a tractable log likelihood lower bound. As a result, no implementation tricks are necessary.

## 3.2 Self-Normalized Importance Sampling (SNIS)

Consider the sampling process defined by self-normalized importance sampling. That is, first sampling a set of $K$ candidate $x_i$s from a proposal distribution $\pi(x_i)$, and then sampling $x$ from the empirical distribution composed of atoms located at each $x_i$ and weighted proportionally to $\exp(-U(x_i))$ (Algorithm 2). In this case, the latent variables $\lambda$ are the locations of the proposal samples $x_1, \ldots, x_K$ (abbreviated $x_{1:K}$) and the index of the selected sample, $i$.

Explicitly, the model is defined by

$$p(x_{1:K}, i) = \left(\prod_{k=1}^{K} \pi(x_k)\right) \frac{\exp(-U(x_i))}{\sum_k \exp(-U(x_k))}, \quad p(x|x_{1:K}, i) = \delta_{x_i}(x),$$

with $\lambda = (x_{1:K}, i)$. We denote the density of the process by $p_{SNIS}(x)$. Choosing $q(\lambda|x) = \frac{1}{K} \delta_{x_i}(x) \prod_{j \neq i} \pi(x_j)$ in Eq. (2), yields

$$\log p_{SNIS}(x) \geq \mathbb{E}_{x_{2:K}} \log \left[ \frac{\pi(x) \exp(-U(x))}{\frac{1}{K}\left(\sum_{j=2}^{K} \exp(-U(x_j)) + \exp(-U(x))\right)} \right]. \tag{3}$$

To summarize, $p_{SNIS}(x)$ can be sampled from exactly and has a tractable lower bound on its log-likelihood. For the same $K$, we expect $p_{SNIS}$ to outperform $p_{TRS}$ because it considers all candidate samples simultaneously instead of sequentially.

---

**Algorithm 2** SNIS$(\pi, U)$ generative process

---

**Require:** Proposal distribution $\pi(x)$ and energy function $U(x)$.
 1: **for** $k = 1, \ldots, K$ **do**
 2:      Sample $x_k \sim \pi(x)$.
 3:      Compute $w(x_k) = \exp(-U(x_k))$.
 4: **end for**
 5: Compute $\hat{Z} = \sum_{k=1}^{K} w(x_k)$
 6: Sample $i \sim \text{Categorical}(w(x_1)/\hat{Z}, \ldots, w(x_K)/\hat{Z})$.
 7: **return** $x = x_i$.

---

As $K \to \infty$, $p_{SNIS}(x)$ becomes proportional to $\pi(x) \exp(-U(x))$. For finite $K$, $p_{SNIS}(x)$ interpolates between the tractable proposal $\pi(x)$ and the energy model $\pi(x) \exp(-U(x))$. Furthermore, Equation (3) is closely connected with the *ranking* NCE loss [34, 46], a popular objective for training energy-based models. In fact, if we consider $\pi(x)$ as our noise distribution $p_N(x)$ and set $U(x) = \log p_N(x) - s(x)$, then up to a constant (in $s$), we recover the ranking NCE loss using the notation from [46]. The ranking NCE loss is motivated by the fact that it is a consistent objective for any $K > 1$ when the true data distribution is in our model family. As a result, it is straightforward to adapt the consistency proof from [46] to our setting. Furthermore, our perspective gives a coherent objective for jointly learning the noise distribution and the energy function and shows that the ranking NCE loss can be viewed as a lower bound on the log likelihood of a well-specified model regardless of whether the true data distribution is in our model family. In addition, we can recover the recently proposed InfoNCE [57] bound on mutual information by using SNIS as the variational distribution in the classic variational bound by Barber and Agakov [4] (see Appendix C for details).

To train the SNIS model, we perform stochastic gradient ascent on Eq. (3) with respect to the parameters of the proposal distribution $\pi$ and the energy function $U$. When the data $x$ are continuous, reparameterization gradients can be used to estimate the gradients to the proposal distribution [61, 38]. When the data are discrete, score function gradient estimators such as REINFORCE [68] or relaxed gradient estimators such as the Gumbel-Softmax [48, 32] can be used.

## 3.3 Hamiltonian importance sampling (HIS)

Simple importance sampling scales poorly with dimensionality, so it is natural to consider more complex samplers with better scaling properties. We evaluated models based on Hamiltonian importance sampling (HIS) [54], which evolve an initial sample under deterministic, discretized

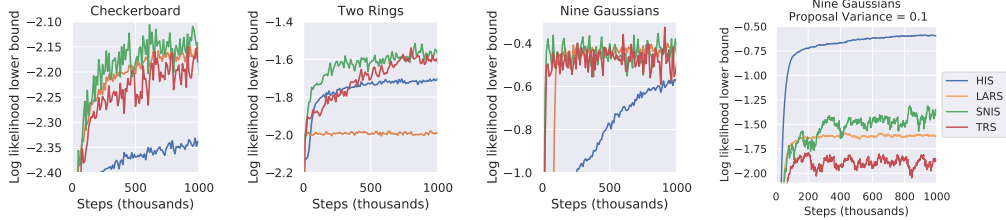

Figure 1: **Performance of LARS, TRS, SNIS, and HIS on synthetic data.** LARS, TRS, and SNIS achieve comparable data log-likelihood lower bounds on the first two synthetic datasets, whereas HIS converges slowly on these low dimensional tasks. The results for LARS on the Nine Gaussians problem match previously-reported results in [5]. We visualize the target and learned densities in Appendix Fig. 2.

Hamiltonian dynamics with a learned energy function. In particular, we sample initial location and momentum variables, and then transition the candidate sample and momentum with leap frog integration steps, changing the temperature at each step (Algorithm 3). While the quality of samples from SNIS are limited by the samples initially produced by the proposal, a model based on HIS updates the positions of the samples directly, potentially allowing for more expressive power. Intuitively, the proposal provides a coarse starting sample which is further refined by gradient optimization on the energy function. When the proposal is already quite strong, drawing additional samples as in SNIS may be advantageous.

In practice, we parameterize the temperature schedule such that $\prod_{t=0}^{T} \alpha_t = 1$. This ensures that the deterministic invertible transform from $(x_0, \rho_0)$ to $(x_T, \rho_T)$ has a Jacobian determinant of 1 (*i.e.*, $p(x_0, \rho_0) = p(x_T, \rho_T)$). Applying Eq. (2) yields a tractable variational objective

$$\log p_{HIS}(x_T) \geq \mathbb{E}_{q(\rho_T|x_T)} \left[ \log \frac{p(x_T, \rho_T)}{q(\rho_T|x_T)} \right] = \mathbb{E}_{q(\rho_T|x_T)} \left[ \log \frac{p(x_0, \rho_0)}{q(\rho_T|x_T)} \right].$$

We jointly optimize $\pi, U, \epsilon, \alpha_{0:T}$, and the variational parameters with stochastic gradient ascent. Goyal et al. [26] propose a similar approach that generates a multi-step trajectory via a learned transition operator.

---

**Algorithm 3** $\text{HIS}(\pi, U, \epsilon, \alpha_{0:T})$ generative process

---

**Require:** Proposal distribution $\pi(x)$, energy function $U(x)$, step size $\epsilon$, temperature schedule $\alpha_0, \ldots, \alpha_T$.
1: Sample $x_0 \sim \pi(x)$ and $\rho_0 \sim \mathcal{N}(0, I)$.
2: $\rho_0 = \alpha_0 \rho_0$
3: **for** $t = 1, \ldots T$ **do**
4: $\quad \rho_t = \rho_{t-1} - \frac{\epsilon}{2} \odot \nabla U(x_{t-1})$
5: $\quad x_t = x_{t-1} + \epsilon \odot \rho_t$
6: $\quad \rho_t = \alpha_t \left( \rho_t - \frac{\epsilon}{2} \odot \nabla U(x_t) \right)$
7: **end for**
8: **return** $x_T$

---

## 4 Experiments

We evaluated the proposed models on a set of synthetic datasets, binarized MNIST [43] and Fashion MNIST [69], and continuous MINST, Fashion MNIST, and CelebA [45]. See Appendix D for details on the datasets, network architectures, and other implementation details. To provide a competitive baseline, we use the recently developed Learned Accept/Reject Sampling (LARS) model [5].

### 4.1 Synthetic data

As a preliminary experiment, we evaluated the methods on modeling synthetic densities: a mixture of 9 equally-weighted Gaussian densities, a checkerboard density with uniform mass distributed in 8

| Method | Static MNIST | Dynamic MNIST | Fashion MNIST |
|---|---|---|---|
| VAE w/ Gaussian prior | $-89.20 \pm 0.08$ | $-84.82 \pm 0.12$ | $-228.70 \pm 0.09$ |
| VAE w/ TRS prior | $-86.81 \pm 0.06$ | $-82.74 \pm 0.10$ | $-227.66 \pm 0.14$ |
| VAE w/ SNIS prior | $-86.28 \pm 0.14$ | $-82.52 \pm 0.03$ | $-227.51 \pm 0.09$ |
| VAE w/ HIS prior | $\mathbf{-86.00 \pm 0.05}$ | $\mathbf{-82.43 \pm 0.05}$ | $-227.63 \pm 0.04$ |
| VAE w/ LARS prior | $-86.53$ | $-83.03$ | $\mathbf{-227.45}^{\dagger}$ |
| ConvHVAE w/ Gaussian prior | $-82.43 \pm 0.07$ | $-81.14 \pm 0.04$ | $-226.39 \pm 0.12$ |
| ConvHVAE w/ TRS prior | $-81.62 \pm 0.03$ | $-80.31 \pm 0.04$ | $-226.04 \pm 0.19$ |
| ConvHVAE w/ SNIS prior | $\mathbf{-81.51 \pm 0.06}$ | $\mathbf{-80.19 \pm 0.07}$ | $\mathbf{-225.83 \pm 0.04}$ |
| ConvHVAE w/ HIS prior | $-81.89 \pm 0.02$ | $-80.51 \pm 0.07$ | $-226.12 \pm 0.13$ |
| ConvHVAE w/LARS prior | $-81.70$ | $-80.30$ | $-225.92$ |
| SNIS w/ VAE proposal | $-87.65 \pm 0.07$ | $-83.43 \pm 0.07$ | $-227.63 \pm 0.06$ |
| SNIS w/ ConvHVAE proposal | $\mathbf{-81.65 \pm 0.05}$ | $\mathbf{-79.91 \pm 0.05}$ | $\mathbf{-225.35 \pm 0.07}$ |
| LARS w/ VAE proposal | — | $-83.63$ | — |

Table 1: **Performance on binarized MNIST and Fashion MNIST.** We report 1000 sample IWAE log-likelihood lower bounds (in nats) computed on the test set. LARS results are copied from [5]. $^{\dagger}$We note that our implementation of the VAE (on which our models are based) underperforms the reported VAE results in [5] on Fashion MNIST.

| Method | MNIST | Fashion MNIST | CelebA |
|---|---|---|---|
| Small VAE | $-1258.81 \pm 0.49$ | $-2467.91 \pm 0.68$ | $-60130.94 \pm 34.15$ |
| LARS w/ small VAE proposal | $-1254.27 \pm 0.62$ | $-2463.71 \pm 0.24$ | $-60116.65 \pm 1.14$ |
| SNIS w/ small VAE proposal | $-1253.67 \pm 0.29$ | $-2463.60 \pm 0.31$ | $-60115.99 \pm 19.75$ |
| HIS w/ small VAE proposal | $\mathbf{-1186.06 \pm 6.12}$ | $\mathbf{-2419.83 \pm 2.47}$ | $\mathbf{-59711.30 \pm 53.08}$ |
| VAE | $-991.46 \pm 0.39$ | $-2242.50 \pm 0.70$ | $-57471.48 \pm 11.65$ |
| LARS w/ VAE proposal | $\mathbf{-987.62 \pm 0.16}$ | $\mathbf{-2236.87 \pm 1.36}$ | $-57488.21 \pm 18.41$ |
| SNIS w/ VAE proposal | $-988.29 \pm 0.20$ | $\mathbf{-2238.04 \pm 0.43}$ | $-57470.42 \pm 6.54$ |
| HIS w/ VAE proposal | $-990.68 \pm 0.41$ | $-2244.66 \pm 1.47$ | $\mathbf{-56643.64 \pm 8.78}$ |
| MAF | $-1027$ | — | — |

Table 2: **Performance on continuous MNIST, Fashion MNIST, and CelebA.** We report 1000 sample IWAE log-likelihood lower bounds (in nats) computed on the test set. As a point of comparison, we include a similar result from a 5 layer Masked Autoregressive Flow distribution [58].

squares, and two concentric rings (Fig. 1 and Appendix Fig. 2 for visualizations). For all methods, we used a unimodal standard Gaussian as the proposal distribution (see Appendix D for further details).

TRS, SNIS, and LARS perform comparably on the Nine Gaussians and Checkerboard datasets. On the Two Rings datasets, despite tuning hyperparameters, we were unable to make LARS learn the density.

On these simple problems, the target density lies in the high probability region of the proposal density, so TRS, SNIS, and LARS only have to reweight the proposal samples appropriately. In high-dimensional problems when the proposal density is mismatched from the target density, however, we expect HIS to outperform TRS, SNIS, and LARS. To test this we ran each algorithm on the Nine Gaussians problem with a Gaussian proposal of mean 0 and variance 0.1 so that there was a significant mismatch in support between the target and proposal densities. The results in the rightmost panel of Fig. 1 show that HIS was almost unaffected by the change in proposal while the other algorithms suffered considerably.

## 4.2 Binarized MNIST and Fashion MNIST

Next, we evaluated the models on binarized MNIST and Fashion MNIST. MNIST digits can be either statically or dynamically binarized — for the statically binarized dataset we used the binarization

from [62], and for the dynamically binarized dataset we sampled images from Bernoulli distributions with probabilities equal to the continuous values of the images in the original MNIST dataset. We dynamically binarize the Fashion MNIST dataset in a similar manner.

First, we used the models as the prior distribution in a Bernoulli observation likelihood VAE. We summarize log-likelihood lower bounds on the test set in Table 1 (referred to as VAE w/ *method* prior). SNIS outperformed LARS on static MNIST and dynamic MNIST even though it used only 1024 samples for training and evaluation, whereas LARS used 1024 samples during training and $10^{10}$ samples for evaluation. As expected due to the similarity between methods, TRS performed comparably to LARS. On all datasets, HIS either outperformed or performed comparably to SNIS. We increased $K$ and $T$ for SNIS and HIS, respectively, and find that performance improves at the cost of additional computation (Appendix Fig. 3). We also used the models as the prior distribution of a convolutional heiarachical VAE (ConvHVAE, following the architecture in [5]). In this case, SNIS outperformed all methods.

Then, we used a VAE as the proposal distribution to SNIS. A limitation of the HIS model is that it requires continuous data, so it cannot be used in this way on the binarized datasets. Initially, we thought that an unbiased, low-variance estimator could be constructed similarly to VIMCO [50], however, this estimator still had high variance. Next, we used the Gumbel Straight-Through estimator [32] to estimate gradients through the discrete samples proposed by the VAE, but found that method performed worse than ignoring those gradients altogether. We suspect that this may be due to bias in the gradients. Thus, for the SNIS model with VAE proposal, we report results on training runs which ignore those gradients. Future work will investigate low-variance, unbiased gradient estimators. In this case, SNIS again outperforms LARS, however, the performance is worse than using SNIS as a prior distribution. Finally, we used a ConvHVAE as the proposal for SNIS and saw performance improvements over both the vanilla ConvHVAE and SNIS with a VAE proposal, demonstrating that our modeling improvements are complementary to improving the proposal distribution.

### 4.3 Continuous MNIST, Fashion MNIST, and CelebA

Finally, we evaluated SNIS and HIS on continuous versions of MNIST, Fashion MNIST, and CelebA (64x64). We use the same preprocessing as in [18]. Briefly, we dequantize pixel values by adding uniform noise, rescale them to $[0, 1]$, and then transform the rescaled pixel values into logit space by $x \to \text{logit}(\lambda + (1 - 2\lambda)x)$, where $\lambda = 10^{-6}$. When we calculate log-likelihoods, we take into account this change of variables.

We speculated that when the proposal is already strong, drawing additional samples as in SNIS may be better than HIS. To test this, we experimented with a smaller VAE as the proposal distribution. As we expected, HIS outperformed SNIS when the proposal was weaker, especially on the more complex datasets, as shown in Table 2.

## 5 Variational Inference with EIMs

To provide a tractable lower bound on the log-likelihood of EIMs, we used the ELBO (Eq. (1)). More generally, this variational lower bound has been used to optimize deep generative models with latent variables following the influential work by Kingma and Welling [38], Rezende et al. [61], and models optimized with this bound have been successfully used to model data such as natural images [60, 39, 11, 27], speech and music time-series [12, 23, 40], and video [2, 29, 17]. Due to the usefulness of such a bound, there has been an intense effort to provide improved bounds [9, 49, 52, 42, 73, 51, 65]. The tightness of the ELBO is determined by the expressiveness of the variational family [74], so it is natural to consider using flexible EIMs as the variational family. As we explain, EIMs provide a conceptual framework to understand many of the recent improvements in variational lower bounds.

In particular, suppose we use a conditional EIM $q(z|x)$ as the variational family (*i.e.*, $q(z|x) = \int q(z, \lambda|x) \, d\lambda$ is the marginalized sampling process). Then, we can use the ELBO lower bound on $\log p(x)$ (Eq. (1)), however, the density of the EIM $q(z|x)$ is intractable. Agakov and Barber [1], Salimans et al. [63], Ranganath et al. [59], Maaløe et al. [47] develop an auxiliary variable

variational bound

$$\mathbb{E}_{q(z|x)}\left[\log\frac{p(x,z)}{q(z|x)}\right] = \mathbb{E}_{q(z,\lambda|x)}\left[\log\frac{p(x,z)r(\lambda|z,x)}{q(z,\lambda|x)}\right] + \mathbb{E}_{q(z|x)}\left[D_{\mathrm{KL}}\left(q(\lambda|z,x)||r(\lambda|z,x)\right]\right.$$

$$\geq \mathbb{E}_{q(z,\lambda|x)}\left[\log\frac{p(x,z)r(\lambda|z,x)}{q(z,\lambda|x)}\right], \tag{4}$$

where $r(\lambda|z,x)$ is a variational distribution meant to model $q(\lambda|z,x)$, and the identity follows from the fact that $q(z|x) = \frac{q(z,\lambda|x)}{q(\lambda|z,x)}$. Similar to Eq. (1), Eq. (4) shows the gap introduced by using $r(\lambda|z,x)$ to deal with the intractability of $q(z|x)$. We can form a lower bound on the original ELBO and thus a lower bound on the log marginal by omitting the positive $D_{\mathrm{KL}}$ term. This provides a tractable lower bound on the log-likelihood using flexible EIMs as the variational family and precisely characterizes the bound gap as the sum of $D_{\mathrm{KL}}$ terms in Eq. (1) and Eq. (4). For different choices of EIM, this bound recovers many of the recently proposed variational lower bounds.

Furthermore, the bound in Eq. (4) is closely related to partition function estimation because $\frac{p(x,z)r(\lambda|z,x)}{q(z,\lambda|x)}$ is an unbiased estimator of $p(x)$ when $z, \lambda \sim q(z,\lambda|x)$. To first order, the bound gap is related to the variance of this partition function estimator (e.g., [49]), which motivates sampling algorithms used in lower variance partition function estimators such as SMC [21] and AIS [53].

## 5.1 Importance Weighted Auto-encoders (IWAE)

To tighten the ELBO without explicitly expanding the variational family, Burda et al. [9] introduced the importance weighted autoencoder (IWAE) bound,

$$\mathbb{E}_{z_{1:K}\sim\prod_i \tilde{q}(z_i|x)}\left[\log\left(\frac{1}{K}\sum_{i=1}^K \frac{p(x,z_i)}{\tilde{q}(z_i|x)}\right)\right] \leq \log p(x). \tag{5}$$

The IWAE bound reduces to the ELBO when $K = 1$, is non-decreasing as $K$ increases, and converges to $\log p(x)$ as $K \to \infty$ under mild conditions [9]. Bachman and Precup [3] introduced the idea of viewing IWAE as auxiliary variable variational inference and Naesseth et al. [52], Cremer et al. [13], Domke and Sheldon [20] formalized the notion.

Consider the variational family defined by the EIM based on SNIS (Algorithm 2). We use a learned, tractable distribution $\tilde{q}(z|x)$ as the proposal $\pi(z|x)$ and set $U(z|x) = \log\tilde{q}(z|x) - \log p(x,z)$ motivated by the fact that $p(z|x) \propto \tilde{q}(z|x)\exp(\log p(x,z) - \log\tilde{q}(z|x))$ is the optimal variational distribution. Similar to the variational distribution used in Section 3.2, setting

$$r(z_{1:K}, i|z, x) = \frac{1}{K}\delta_{z_i}(z)\prod_{j\neq i}\tilde{q}(z_j|x) \tag{6}$$

yields the IWAE bound Eq. (5) when plugged into to Eq. (4) (see Appendix A for details).

From Eq. (4), it is clear that IWAE is a lower bound on the standard ELBO for the EIM $q(z|x)$ and the gap is due to $D_{\mathrm{KL}}(q(z_{1:K}, i|z, x)||r(z_{1:K}, i|z, x))$. The choice of $r(z_{1:K}, i|z, x)$ in Eq. (6) was for convenience and is suboptimal. The optimal choice of $r$ is

$$q(z_{1:K}, i|z, x) = q(i|z, x)q(z_{1:K}|i, z, x) = \frac{1}{K}\delta_{z_i}(z)q(z_{-i}|i, z, x).$$

Compared to the optimal choice, Eq. (6) makes the approximation $q(z_{-i}|i, z, x) \approx \prod_{j\neq i}\tilde{q}(z_j|x)$ which ignores the influence of $z$ on $z_{-i}$ and the fact that $z_{-i}$ are not independent given $z$. A simple extension could be to learn a factored variational distribution conditional on $z$: $r(z_{1:k}, i|z, x) = \frac{1}{K}\delta_{z_i}(z)\prod_{j\neq i}r(z_j|z, x)$. Learning such an $r$ could improve the tightness of the bound, and we leave exploring this to future work.

## 5.2 Semi-implicit variational inference

As a way of increasing the flexibility of the variational family, Yin and Zhou [73] introduce the idea of semi-implicit variational families. That is they define an implicit distribution $q(\lambda|x)$ by transforming a random variable $\epsilon \sim q(\epsilon|x)$ with a differentiable deterministic transformation (*i.e.*,

$\lambda = g(\epsilon, x)$). However, Sobolev and Vetrov [65] keenly note that $q(z, \lambda | x) = q(z | \lambda, x) q(\lambda | x)$ can be equivalently written as $q(z | \epsilon, x) q(\epsilon | x)$ with two explicit distributions. As a result, semi-implicit variational inference is simply auxiliary variable variational inference by another name.

Additionally, Yin and Zhou [73] provide a multi-sample lower bound on the log likelihood which is generally applicable to auxiliary variable variational inference.

$$\log p(x) \geq \mathbb{E}_{q(\lambda_{1:K-1}|x)q(z,\lambda|x)} \left[ \log \frac{p(x,z)}{\frac{1}{K} \left( q(z|\lambda,x) + \sum_i q(z|\lambda_i,x) \right)} \right] \tag{7}$$

We can interpret this bound as using an EIM for $r(\lambda | z, x)$ in Eq. (4). Generally, if we introduce additional auxiliary random variables $\gamma$ into $r(\lambda, \gamma | z, x)$, we can tractably bound the objective

$$\mathbb{E}_{q(z,\lambda|x)} \left[ \log \frac{p(x,z)r(\lambda|z,x)}{q(z,\lambda|x)} \right] \geq \mathbb{E}_{q(z,\lambda|x)s(\gamma|z,\lambda,x)} \left[ \log \frac{p(x,z)r(\lambda,\gamma|z,x)}{q(z,\lambda|x)s(\gamma|z,\lambda,x)} \right], \tag{8}$$

where $s(\gamma | z, \lambda, x)$ is a variational distribution. Analogously to the previous section, we set $r(\lambda | z, x)$ as an EIM based on the self-normalized importance sampling process with proposal $q(\lambda | x)$ and $U(\lambda | x, z) = -\log q(z | \lambda, x)$. If we choose

$$s(\lambda_{1:K}, i | z, \lambda, x) = \frac{1}{K} \delta_{\lambda_i}(\lambda) \prod_{j \neq i} q(\lambda_j | x),$$

with $\gamma = (\lambda_{1:K}, i)$, then Eq. 8 recovers the bound in [73] (see Appendix B for details). In a similar manner, we can continue to recursively augment the variational distribution $s$ (*i.e.*, add auxiliary latent variables to $s$).

This view reveals that the multi-sample bound from [73] is simply one approach to choosing a flexible variational $r(\lambda | z, x)$. Alternatively, Ranganath et al. [59] use a learned variational $r(\lambda | z, x)$. It is unclear when drawing additional samples is preferable to learning a more complex variational distribution. Furthermore, the two approaches can be combined by using a learned proposal $r(\lambda_i | z, x)$ instead of $q(\lambda_i | x)$, which results in a bound described in [65].

### 5.3 Additional Bounds

Finally, we can also use the self-normalized importance sampling procedure to extend a proposal family $q(z, \lambda | x)$ to a larger family (instead of solely extending $r(\lambda | z, x)$) [65]. Self-normalized importance sampling is a particular choice of taking a proposal distribution and moving it closer to a target. Hamiltonian Monte Carlo [55] is another choice which can also be embedded in this framework as done by [63, 10]. Similarly, SMC can be used as a sampling procedure in an EIM and when used as the variational family, it succinctly derives variational SMC [49, 52, 42] without any instance specific tricks. In this way, more elaborate variational bounds can be constructed by specific choices of EIMs without additional derivation.

## 6 Discussion

We proposed a flexible, yet tractable family of distributions by treating the approximate sampling procedure of energy-based models as the model of interest, referring to them as *energy-inspired models*. The proposed EIMs bridge the gap between learning and inference in EBMs. We explore three instantiations of EIMs induced by truncated rejection sampling, self-normalized importance sampling, and Hamiltonian importance sampling and we demonstrate comparably or stronger performance than recently proposed generative models. The results presented in this paper use simple architectures on relatively small datasets. Future work will scale up both the architectures and size of the datasets.

Interestingly, as a by-product, exploiting the EIMs to define the variational family provides a unifying framework for recent improvements in variational bounds, which simplifies existing derivations, reveals potentially suboptimal choices, and suggests ways to form novel bounds.

Concurrently, Nijkamp et al. [56] investigated a similar model to our models based on HIS, although the training algorithm was different. Combining insights from their study with our approach is a promising future direction.

**Acknowledgments**

We thank Ben Poole, Abhishek Kumar, and Diederick Kingma for helpful comments. We thank Matthias Bauer for answering implementation questions about LARS.

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
