[Supplementary Material]

# Appendices

## A   IWAE bound as AVVI with an EIM

We provide a proof sketch that the IWAE bound can be interpreted as auxiliary variable variational inference with an EIM. Recall the auxiliary variable variational inference bound (Eq. (1) and Eq. (4)),

$$\log p(x) \geq \mathbb{E}_{q(z|x)}\left[\log \frac{p(x,z)}{q(z|x)}\right] \geq \mathbb{E}_{q(z,\lambda|x)}\left[\log \frac{p(x,z)r(\lambda|z,x)}{q(z,\lambda|x)}\right]. \tag{9}$$

Let $q$ be an EIM based on SNIS with proposal $\tilde{q}(z|x)$ and energy function $U(z|x) = \log \tilde{q}(z|x) - \log p(x,z)$ and $r$ be

$$r(z_{1:K}, i|z,x) = \frac{1}{K}\delta_{z_i}(z)\prod_{j \neq i}\tilde{q}(z_j|x). \tag{10}$$

Then, plugging Eq. (10) into Eq. (9) with $\lambda = (z_{1:K}, i)$ gives

$$\log p(x) \geq \mathbb{E}_{q(z,\lambda|x)}\left[\log \frac{p(x,z)r(\lambda|z,x)}{q(z,\lambda|x)}\right] = \mathbb{E}_{q(\lambda|x)}\left[\log \frac{p(x,z_i)\frac{1}{K}\prod_{j\neq i}\tilde{q}(z_j|x)}{\frac{w_i}{\sum_j w_j}\prod_j \tilde{q}(z_j|x)}\right]$$

$$= \mathbb{E}_{q(z_{1:K}, i|x)}\left[\log \frac{1}{K}\sum_j w_j\right] = \mathbb{E}_{\prod_j \tilde{q}(z_j)}\left[\log \frac{1}{K}\sum_j w_j\right],$$

which is the IWAE bound.

## B   Semi-implicit Variational Inference Bound

$$\log p(x) \geq \mathbb{E}_{q(z,\lambda|x)s(\gamma|z,\lambda,x)}\left[\log \frac{p(x,z)r(\lambda, \gamma|z,x)}{q(z,\lambda|x)s(\gamma|z,\lambda,x)}\right]$$

$$= \frac{1}{K}\sum_i \mathbb{E}_{q(z,\lambda|x)s(\lambda_{1:K}|i,x)}\left[\log \frac{p(x,z)r(\lambda, \gamma|z,x)}{q(z,\lambda|x)s(\gamma|z,\lambda,x)}\right]$$

$$= \frac{1}{K}\sum_i \mathbb{E}\left[\log \frac{p(x,z)q(\lambda|x)q(z|\lambda,x)}{q(z,\lambda|x)\frac{1}{K}\left(\sum_{j\neq i}w(z_j) + q(z|\lambda,x)\right)}\right]$$

$$= \frac{1}{K}\sum_i \mathbb{E}\left[\log \frac{p(x,z)}{\frac{1}{K}\left(\sum_{j\neq i}q(z|\lambda_j,x) + q(z|\lambda,x)\right)}\right],$$

which is equivalent to the multi-sample bound from [73].

## C   Connection with CPC

Starting from the well-known variational bound on mutual information due to Barber and Agakov [4]

$$I(X,Y) = \mathbb{E}_{p(x,y)}\left[\log \frac{p(x,y)}{p(x)p(y)}\right] \geq \mathbb{E}_{p(x,y)}\left[\log \frac{q(x|y)}{p(x)}\right]$$

for a variational distribution $q(x|y)$, we can use the self-normalized importance sampling distribution and choose the proposal to be $p(x)$ (*i.e.*, $p_{SNIS(p,U)}$). Applying the bound in Eq. (3), we have

$$I(X,Y) \geq \mathbb{E}_{p(x,y)}\left[\log \frac{p_{SNIS(p,U)}(x|y)}{p(x)}\right]$$

$$\geq \mathbb{E}_{p(x,y)}\mathbb{E}_{x_{2:K}} \log \left[\frac{\exp\left(-U(x,y)\right)}{\frac{1}{K}\left(\sum_j \exp\left(-U(x_j,y)\right) + \exp\left(-U(x,y)\right)\right)}\right].$$

This recovers the CPC bound and proves that it is indeed a lower bound on mutual information whereas the heuristic justification in the original paper relied on unnecessary approximations.

# D Implementation Details

## D.1 Synthetic data

All methods used a fixed 2-D $\mathcal{N}(0, 1)$ proposal distribution and a learned acceptance/energy function $U(x)$ parameterized by a neural network with 2 hidden layers of size 20 and tanh activations. For SNIS and LARS, the number of proposal samples drawn, $K$, was set to 1024 and for HIS $T = 5$. We used batch sizes of 128 and ADAM [37] with a learning rate of $3 \times 10^{-4}$ to fit the models. For evaluation, we report the IWAE bound with 1000 samples for HIS and SNIS. For LARS there is no equivalent to the IWAE bound, so we instead estimate the normalizing constant with 1000 samples.

The nine Gaussians density is a mixture of nine equally-weighted 2D Gaussians with variance 0.01 and means $(x, y) \in \{-1, 0, 1\}^2$. The checkerboard density places equal mass on the squares $\{[0, 0.25], [0.5, 0.75]\}^2$ and, $\{[0.25, 0.5], [0.75, 1.0]\}^2$. The two rings density is defined as

$$p(x, y) \propto \mathcal{N}(\sqrt{x^2 + y^2}; 0.6, 0.1) + \mathcal{N}(\sqrt{x^2 + y^2}; 1.3, 0.1).$$

## D.2 Binarized MNIST and Fashion MNIST

We chose hyperparameters to match the MNIST experiments in Bauer and Mnih [5]. Specifically, we parameterized the energy function by a neural network with two hidden layers of size 100 and tanh activations, and parameterized the VAE observation model by neural networks with two layers of 300 units and tanh activations. The latent spaces of the VAEs were 50-dimensional, SNIS's $K$ was set to 1024, and HIS's $T$ was set to 5. We also linearly annealed the weight of the KL term in the ELBO from 0 to 1 over the first $1 \times 10^5$ steps and dropped the learning rate from $3 \times 10^{-4}$ to $1 \times 10^{-4}$ on step $1 \times 10^6$. All models were trained with ADAM [37].

## D.3 Continuous MNIST, Fashion MNIST, and CelebA

For the small VAE, we parameterized the VAE observation model neural networks with a single layer of 20 units and tanh activations. The latent spaces of the small VAEs were 10-dimensional. In these experiments, SNIS's $K$ was set to 128, and HIS's $T$ was set to 5. We also dropped the learning rate from $3 \times 10^{-4}$ to $1 \times 10^{-4}$ on step $1 \times 10^6$. All models were trained with ADAM [37].

Figure 2: **Target and Learned Densities for Synthetic Examples.** The visualizations for TRS, SNIS, and HIS are approximated by drawing samples.

Figure 3: **Performance while varying $K$ and $T$ for VAE w/ SNIS prior and VAE w/ HIS prior.** As expected, in both cases, increasing $K$ or $T$ improves performance at the cost of additional computation.