[Reviews · NeurIPS 2019]

Reviewer 1



The paper was a pleasure to read. The contribution was stated clearly and the literature reviewed extensively, with interesting links being drawn to multiple techniques. To the best of my knowledge this general view of auxiliary variables, and of embedding samplers in latent space to obtain generative models is a novel contribution (at least in this generalised form, special cases were known, cf. HMC and SMC) that has both a didactic and, potentially, a practical use. Unfortunately the empirical results seem inconclusive, and no fundamentally new algorithm is put forward: in the paper well-known sampling methods (SNIS and HIS) hitherto used for inference have been applied as generative models. This alone does not, in my opinion, constitute a sufficiently original contribution, especially since the resulting methods marginally under or over-perform compared with existing methods. Despite this, given the useful insights and unification this paper provides, I would recommend it for acceptance, provided the points below are addressed. ########### POST REBUTTAL ############### I'd like to thank the authors for addressing the reviewers' concerns. In particular for running the experiments requested in the short amount of time available. The consensus that emerged from discussing with the other reviewers, however, is that the main contribution of the paper is of a "didactic" nature, rather than a practical one. I must admit fault for not emphasising this point enough before the rebuttal, but I feel that while this work could potentially be very useful and exciting, this potential has not yet been borne out. I think the main way to address this would be to show how this framework allows us to study various choices of conditional distributions on the auxiliary variables and to classify the resulting tightness of the bound (for fixed model parameters) as well as the quality of the resulting samples (or predictive accuracy/calibration in the case of a predictive downstream task). I do remain of the opinion that this paper is well written and deserves to be shared. So I keep my score a 6 and recommend that this paper be accepted.

Reviewer 2



Summary: The paper begins by introducing auxiliary variable variational inference (AVVI) by starting from the usual ELBO and introducing an additional variable lambda and distribution r(lambda | x,z), which is meant to model q(lambda|z,x). Based on previous work, they show that using a self-normalized importance sampling process in AVVI, they recover IWAE. They note that the original IWAE bound used is suboptimal and propose a possible improvement. The paper continues by embedding the importance sampler and one based on Hamiltonian MC into a generative model (VAE specifically). The resulting algorithms are called Self-Normalized Importance Sampling (SNIS) and Hamiltonian Importance Sampling (HIS). The algorithms are evaluated on MNIST (binarized in two different ways) and Fashion MNIST (also binarized), and continuous versions of MNIST, FashionMNIST and CelebA. In all their experiments they outperform their baseline (a standard VAE) and Learned Accept/Reject Sampling (LARS). Discussion: The paper is reasonably clearly written, with a lot of attention given to the derivation of AVVI, SNIS and HIS. Section 3.2 seems to not directly relate to other parts of the paper, while 3.3 is very dense. The introduction can use more focus, e.g. by defining auxiliary latent variables, giving more context on bound tightness (surely it's also dependent on the model fit), more context on why the relationship between different tighter variational lower bounds is "unclear". The authors do a tremendous job of citing previous work, calling out specific papers for specific incremental improvements. The experimental setup can use more context. The authors seem to have followed the phrasing and set up of LARS, but without consulting that paper it's unclear what "SNIS w/ VAE proposal" means (line 240 is only explanation it seems). The main problems with the paper are experimental evaluation and novelty. On experimental evaluation, the first unclarity is why HIS performs so much worse in the synthetic setup. The authors mention they expect HIS to be comparably better when the proposal density is mismatched, but that does not explain why HIS is not great on synthetic data. Table 1 show that the proposed methods improve upon both the baseline and LARS on MNIST, and perform slightly worse on FashionMNIST (although the baseline of LARS is 228.68 vs 229.02 in this paper, so the methods perform roughly the same). Table 2 shows results, but with few comparisons. It would be interesting to see a comparison with at least LARS. Also given that the proposed methods can be bolted on previous methods that tighten variational bounds, it would be interesting to see a more thorough comparison. Example questions for the thorough comparison: - When is HIS vs HVAE vs HIS+HVAE a good choice? How should a user trade off computation? - If we improve our proposal (e.g. by using a stronger model like convHVAE), will HIS/SNIS still provide benefits? -- Rebuttal -- The authors gave convincing answers to the above questions and I update my score to a 6. The main drawback is still the limited improvements for a considerable increase in method complexity.

Reviewer 3



The paper studies the self-normalized importance sampling in two scenarios. First, the authors show that both IWAE and semi-implicit variational inference can be formulated as a special case of auxiliary variable variational inference with self-normalized importance sampling. Second, the authors proposed a generative model based on the self-normalized importance sampling, in which the samples of the posterior distribution can be approximated by drawing from a reweighted distribution based on the samples from the prior distribution. To overcome the limitation of directly reweighting the samples from the prior distribution, the authors proposed to use Hamiltonian importance sampling. In the experiments, the proposed generative model can compared with standard VAE. The main concerns of the paper: 1. The paper is a combination of two separate topics. Although both topics are about self-normalized importance sampling, they are used in different ways: (1) used as variational posterior for variational inference (2) used to form a generative model. This makes the paper fragmented and hard to read. 2. Most of the findings exist in the literature in the same or varied form, apart from the re-interpretation of semi-implicit variational inference. The authors need to better highlight the original findings vs the existing understanding in the literature. Originality: Most of the findings exist in the literature in the same or varied form. Quality, clarity: The paper is fragmented and many technical details are omitted, e.g., what is the formulation of the energy functions used in the experiments and why the energy function formulation is better than a standard VAE. Significance: If the presentation of the paper is improved, this work can help to improve the understanding about self-normalized importance sampling in the community. ------ In rebuttla, the authors provide a much clearer explanation of the contributions. The main contribution of the paper is a specific auxiliary variable variational inference formulation that unifies a few previous works. It would be nice if the authors can show how the proposed formulation can be used to improve existing methods such as IWAE in details.

[Author Response · NeurIPS 2019]

We thank the reviewers for their suggestions. We have revised the text for clarity, added experiment details so that the paper is self-contained, and added the following comparisons requested by reviewers: 1) a comparison of SNIS, HIS, and LARS on the Continuous MNIST, Fashion MNIST, and CelebA datasets, 2) a comparison of performance as $K$ and $T$ are varied, and 3) experiments which use HIS and SNIS as the prior for a convolutional hierarchical VAE (ConvHVAE) as well as using the ConvHVAE as a proposal for SNIS. These experiments confirm that SNIS and HIS outperform or perform comparably to LARS while optimizing a proper lower bound and being simpler to implement. We include a subset of these results in Tables 1 and 2. We have also added samples from each model to the Appendix to allow for qualitative comparisons.

**Contributions.** The paper characterizes the bound gap for Monte Carlo Objectives, explicitly reveals the connection with auxiliary variable variational inference, and derives a novel class of models that balance tractability with the inductive biases of energy-based models. Furthermore, as **R1** notes, we draw links between disparate techniques and unify many existing approaches in a common framework.

| Method | Static | Dynamic | Fashion |
|---|---|---|---|
| ConvHVAE | −82.43 | −81.14 | −226.39 |
| ConvHVAE w/ SNIS prior | **−81.51** | **−80.19** | **−225.83** |
| ConvHVAE w/ HIS prior | −81.89 | −80.52 | −226.15 |
| ConvHVAE w/ LARS prior | −81.70 | −80.30 | −225.92 |
| SNIS w/ ConvHVAE prop. | **−81.65** | **−79.93** | **−225.53** |
| SNIS w/ VAE prop. | −87.65 | −83.43 | −227.63 |
| LARS w/ VAE prop. | — | −83.63 | — |

**Table 1: Discrete MNIST datasets.**

| Method | MNIST | Fashion |
|---|---|---|
| SNIS w/Small VAE prop. | −1254.77 | −2462.60 |
| HIS w/Small VAE prop. | **−1207.88** | **−2449.48** |
| LARS w/Small VAE prop. | −1256.83 | −2464.65 |
| SNIS w/VAE prop. | −1000.83 | **−2244.06** |
| HIS w/VAE prop. | **−996.32** | −2250.19 |
| LARS w/ VAE prop. | −1005.20 | −2263.46 |

**Table 2: Continuous datasets.**

**Figure 1: Varying $T$ & $K$.**

**R1: Compare with LARS on the continuous datasets.** We have added this comparison and find that LARS underperforms SNIS and HIS (Table 2 and similar results on CelebA).

**R1: Varying $T$ and $K$.** We have added this comparison and find that increasing $K$ or $T$ improves performance at the cost of more computation (Fig. 1).

**R1: Can existing lower bounds can be improved and a more general analysis of bound tightness?** We agree that it would be interesting to experimentally examine this; however, due to space constraints, we chose to focus our experiments on the proposed models. Theoretically, we generically characterize the bound gap in terms of KL divergences (Eqs. 1 & 2). While the bound gap was known for specific cases (e.g., IWAE), we can use the general result to characterize the bound gap of VSMC, the Hamiltonian VAE, and semi-implicit VI. To first order, the bound gap is related to the variance of the partition function estimator, which motivates using lower variance estimators. We now explain this in the main text.

**R2, R3: Clarity.** We have rewritten the introduction to focus on the key concepts that are used later and to bridge the two halves of the paper. Section 3.2 was intended as a complex example of a method falling into our framework. We have rewritten the section to more clearly connect it back to the central story. We have reworked the experimental sections so that the setup is clear without having to read LARS.

**R2: Why does HIS perform worse on synthetic data?** Our implementation of HIS implicitly uses $K = 1$ and can be extended to $K > 1$ by drawing additional samples, reweighting, and sampling. Increasing $K$ or $T$ improves the performance of HIS on the synthetic data. To verify our claims about density mismatch, we reran the synthetic experiments where the proposal distribution has smaller variance and found that HIS outperforms the other methods. We now include these comparisons in the Appendix.

**R2: HIS vs HVAE vs HIS+HVAE.** Our experiments show that they provide complementary improvements. SNIS and HIS improve the performance of both the VAE and ConvHVAE when used as the prior distribution. Because the latent space is small (50-dimensional), the additional computation cost of SNIS or HIS is small.

**R2: With stronger proposals, do HIS/SNIS still provide benefits?** Yes, we now include experiments with a ConvHVAE as the proposal and show that SNIS continues to improve performance (Table 1).

**R3: The formulation of energy functions is not mentioned.** We have moved the details from Appendix D to the main text.

**R3: Benefit of the energy function formulation?** Asymptotically, neither is superior, however, in practice, the energy function exploits different inductive biases than the VAE. Instead of directly specifying a generative distribution, it determines the distribution by scoring images. As we show in the experiments, the VAE and the energy function formulation are complementary and combining them produces the best results. We have added this intuition to the main text.

[Meta-Review · NeurIPS 2019]

The reviewers had an in depth discussion about this submission, suggesting a weak accept based on the paper's didactic contributions, but also commenting extensively on the limitations of the paper. I've decided to recommend to accept this submission, but please incorporate the reviewers' suggested improvements into your camera ready version of the paper, and in particular: * the reviewers' request for a practical discussion was ignored * demonstrate practical implications with the unified formulation (eg "A simple extension could be [..] we leave exploring this to future work") * demonstrate briefly useful extensions such as suggesting a better auxiliary variable distribution for IWAE (which are only mentioned as a possibility like in Line 108)